# Real-World Outcome and Prognostic Factors in MDS Patients Treated with Azacitidine—A Retrospective Analysis

**DOI:** 10.3390/cancers16071333

**Published:** 2024-03-29

**Authors:** Kamil Wiśniewski, Katarzyna Pruszczyk-Matusiak, Bartosz Puła, Ewa Lech-Marańda, Joanna Góra-Tybor

**Affiliations:** 1Department of Hematology, Institute of Hematology and Transfusion Medicine, 02-776 Warsaw, Polandewamaranda@wp.pl (E.L.-M.); 2Department of Hematology, Medical University of Lodz, 93-513 Lodz, Poland

**Keywords:** azacitidine, myelodysplastic syndrome, prognostic factors, real-world, survival

## Abstract

**Simple Summary:**

Azacitidine (AZA) is an essential drug in the treatment of myelodysplastic syndromes (MDS) that has made it possible to extend patients’ survival and improve their quality of life. Unfortunately, despite the widespread use of AZA, its prognostic factors for response still remain unknown. Here, we retrospectively analyzed the efficacy and safety of AZA therapy in 79 patients with MDS in a real-life setting. Furthermore, we provided some potential biomarkers of response and survival. The study confirmed that the achievement of response to AZA is not mandatory for obtaining a survival benefit in patients with MDS. Unfavorable cytogenetic risk was determined to be the most negative prognostic factor for both response and survival. Moreover, older age, a complex or monosomal karyotype, higher IPSS or IPSS-R risk and a higher level of serum ferritin level were associated with significantly shorter survival.

**Abstract:**

Azacitidine (AZA) is recognized as a vital drug used in the therapy of myelodysplastic syndromes (MDS) due to its beneficial effect on survival and quality of life. Nevertheless, many patients fail to respond to AZA treatment, as prognostic factors still are not identified. The present retrospective analysis included 79 patients with MDS treated with AZA as first-line therapy in a real-life setting. The percentage of patients with good, intermediate, and poor cytogenetics was 46.8%, 11.4%, and 34.2%, respectively. The overall response rate (complete remission [CR], partial remission [PR], and hematological improvement [HI]) was 24%. The CR, PR, and HI rates were 13.9%, 2.5%, and 7.6%, respectively. Stable disease (SD) was documented in 40.5% of patients. The median overall survival (OS) and progression-free survival (PFS) were 17.6 and 14.96 months, respectively. Patients with ORR and SD had a significantly longer median OS (23.8 vs. 5.7 months, *p* = 0.0005) and PFS (19.8 vs. 3.5 months, *p* < 0.001) compared to patients who did not respond to AZA. In univariate analysis, only an unfavorable cytogenetic group was a prognostic factor of a lower response rate (*p* = 0.03). In a multivariate model, older age (*p* = 0.047), higher IPSS (International Prognostic Scoring System) risk (*p* = 0.014), and higher IPSS-R cytogenetic risk (*p* = 0.004) were independent factors of shorter OS. Independent prognostic factors for shorter PFS were age (*p* = 0.001), IPSS risk (*p* = 0.02), IPSS cytogenetic risk (*p* = 0.002), and serum ferritin level (*p* = 0.008). The safety profile of AZA was predictable and consistent with previous studies. In conclusion, our study confirms the efficacy and safety of AZA in a real-world population and identifies potential biomarkers for response and survival.

## 1. Introduction

Myelodysplastic syndromes (MDS) represent a diverse group of hematologic cancers that are characterized by ineffective hematopoiesis, leading to blood cytopenia and a high risk of progression to acute myeloid leukemia (AML). MDS is mostly a disease of elderly people, with a median age at diagnosis of 70 years [1]. Despite substantial improvement in diagnostic methods and therapeutic strategies, the prognosis and survival outcomes of MDS patients remain poor, particularly in the case of elderly patients ineligible for intensive chemotherapy. Azacitidine (AZA) treatment has become the standard of care for this group of patients since they benefit from improvements in overall survival, quality of life, and less toxicity as compared with conventional chemotherapy [1,2,3,4,5]. 

Nevertheless, nearly half of patients fail to respond to AZA treatment, and most of the responders are likely to suffer from disease progression [6,7,8,9]. Differential clinical outcome is caused by a remarkable diversity of clinical and genetic features in MDS. Unfortunately, no clear prognostic factors for response to AZA treatment have been identified so far. Recent studies have provided limited and conflicting evidence on a few potential biomarkers associated with clinical response [8,9,10,11,12,13,14,15,16,17,18,19,20,21,22]. Moreover, the most recent data available come from clinical trial populations. There is an unmet medical need to determine predictive factors in response to AZA therapy based on studies performed in a real-world clinical practice setting. 

The aim of the present study was to evaluate the efficacy and toxicity of AZA treatment for “real life” MDS patients. The paper provides an analysis of prognostic factors associated with clinical response and survival in patients treated with AZA.

## 2. Patients and Methods

### 2.1. Patients

This retrospective analysis included data collected from MDS patients treated with AZA as first-line therapy at the Institute of Hematology and Transfusion Medicine in Warsaw (Poland) (IHT) during the period between 2013 and 2018. The study included only patients who had received at least one complete course of AZA. The diagnosis of MDS was established according to the International Working Group (IWG) 2006 criteria [23]. Risk stratification was established according to the International Prognostic Scoring System (IPSS) and revised IPSS (IPSS-R) [24,25].

The study received a positive ethical evaluation from the IHT Bioethics Committee on 7 March 2019.

### 2.2. Treatment

Patients received a standard dose of AZA, 75 mg/m^2^ subcutaneously for 7 days, every 4 weeks. Both dosing regimens (7 consecutive days and 5-2-2 days) were allowed. Schedule modifications and dose adjustment were permitted due to hematological toxicity according to common clinical practice. AZA treatment was continued until disease progression, unacceptable toxicity, or death.

### 2.3. Data Collection

Baseline patient and disease characteristics were collected retrospectively. The following data were gathered: age, sex, Eastern Cooperative Oncology Group (ECOG) performance status, body mass index (BMI), hematopoietic cell transplantation comorbidity index (HCT-CI), disease subtype according to World Health Organization (WHO) 2016 classification [26], IPSS and IPSS-R risk score, cytogenetic abnormalities, transfusion dependence, percentage of bone marrow (BM) blasts, presence of peripheral blasts, white blood count (WBC), absolute neutrophil count (ANC), platelet count (PLT), hemoglobin level (HGB), lactate dehydrogenase level (LDH) and serum ferritin (SF) level. Cytogenetic testing included karyotype analysis and cytogenetic risk assessment based on the IPSS and IPSS-R. Transfusion dependence was defined as requiring at least one unit of red blood cell (RBC) or platelets before start of AZA treatment.

### 2.4. Response and Outcome Criteria 

The response to treatment was determined using the IWG 2006 criteria [23]. The overall response rate (ORR) was defined as the percentage of patients who achieved complete remission (CR), partial remission (PR), and hematological improvement (HI). Only patients without at least PR were considered eligible for an assessment of HI. Transfusion independence (TI) was defined as the absence of RBC or PLT transfusions for at least 8 weeks. Response evaluation was performed after 3 to 6 cycles using blood count, peripheral smear, and BM aspiration. Response in patients who had not received at least three courses of AZA was evaluated by blood count and peripheral smear. Response to treatment was defined as the best at any point during AZA therapy. Overall survival (OS) was defined from AZA initiation to the date of death or last observation. Progression-free survival (PFS) was measured from the onset of AZA until the date of progression, death, or last observation.

### 2.5. Safety Assessment

The safety assessment included analysis of hematological and non-hematological toxicity. Adverse events (AE) were categorized on the basis of the Common Terminology Criteria for Adverse Events (CTCAE), version 5.0 [27]. Severe adverse events (SAE) were determined as AEs of Grade ≥ 3 based on the CTCAE. Hematological toxicity (anemia, thrombocytopenia, neutropenia) was defined as either the occurrence of cytopenia or the worsening of existing cytopenia (≥1 grade) during AZA treatment. Analysis of infectious toxicity was based on the type, severity, and number of infectious episodes during AZA therapy. Furthermore, the study evaluated schedule modifications and dose adjustments caused by toxicity during AZA treatment. 

### 2.6. Statistical Analysis

Patients with MDS were evaluated in terms of the impact of clinical characteristics on response to AZA and patient outcome (OS and PFS). Clinical characteristics were determined on the basis of frequency (percentage) and median (range) for categorical and continuous variables, respectively. The Mann–Whitney U test and Pearson’s chi-square test were used to analyze continuous and discrete variables, respectively. Survival curves were estimated using the Kaplan–Meier method and compared with the log-rank test. A *p*-value < 0.05 was considered significant. Logistic regression analysis aimed at determining independent factors related to the OS and PFS. Multivariate analysis was conducted with the Cox proportional hazard model. The multivariate model included only the most significant variables with a *p*-value < 0.1. 

All statistical analyses were conducted using Statistica version 13.3 (StatSoft, Dell, Round Rock, TX, USA), Graph Pad Prism version 9.0 for Windows (GraphPad Software, San Diego, CA, USA), and SAS software (SAS Institute Inc., Cary, NC, USA).

## 3. Results

### 3.1. Patient Characteristics

The analysis included a total of 79 patients with MDS who received at least one complete cycle of AZA at our institution during the entire study period. There were 45 (57%) males and 34 (43%) females with a median age of 69 years (range 42–89). The diagnoses according to WHO 2016 classification included 66 patients (83.5%) with MDS with excess blasts (MDS-EB); 6 (7.6%) with multilineage dysplasia MDS (MDS-MLD); 5 (6.3%) with therapy-related MDS (t-MDS); 1 (1.3%) with ring sideroblasts MDS (MDS-RS), and 1 (1.3%) with MDS 5q−. According to the IPSS, 20 (25.3%), 45 (57%), and 14 (17.7%) patients belonged to the intermediate-1, intermediate-2, and high-risk groups, respectively. In terms of IPSS-R, 16 (20.3%), 29 (36.7%), and 34 (43%) patients were classified as intermediate, high, and very high risk, respectively. Cytogenetics results were obtained in most patients (92.4%, n = 73). Complex and monosomal karyotypes were identified in 22 (27.9%) and 17 (21.5%) patients, respectively. Most of the patients (70.9%, n = 56) were transfusion-dependent. The baseline patient characteristics are presented in Table 1.

### 3.2. Response and Outcome

Patients received a median of six AZA cycles (range 1–34). Among all patients, 50 (63.3%) and 27 (34.8%) were administrated at least 6 and 12 cycles of AZA, respectively. The median time from MDS diagnosis to AZA therapy initiation was 1 month (range: 0–69) with the median duration of treatment reported as 6 months (range: 1–34). ORR reached 24%, including 13.9% CR and 2.5% PR rates. HI and transfusion independence were documented in 7.6% and 25% of patients, respectively. Stable disease (SD) was observed in 40.5% of patients. The median number of AZA cycles needed to achieve the best response was six (range: 1–12). Details of AZA treatment are shown in Table 2. At the data cut-off point (03/2021), ten patients (12.7%) remained under observation and three (3.8%) were continuing AZA treatment. A total of 24 (30.4%) patients received a second line of therapy because of progressive disease. Allogeneic hematopoietic stem cell transplantation (allo-HSCT) was performed in 10 patients (12.7%). 

### 3.3. Safety

Hematological toxicity was reported in 28 (35.4%) patients with neutropenia, thrombocytopenia, and anemia observed in 20.3%, 11.4%, and 3.8% of the patients, respectively. The incidence of grade 3–4 cytopenia was 31.2%. Non-hematological toxicity occurred in 38% of patients. The most common non-hematological adverse events were injection site reactions (21.5%) with mild and transient courses. Grade 3–4 non-hematological toxicity was reported in only seven (8.9%) patients with four cases of bleeding events. More detailed information is shown in Table 3. Fifty-five (69.6%) patients experienced at least one infectious episode. The most common types of infection were pneumonia (29.1%), upper respiratory tract infection (19%), neutropenic fever (17.7%), and skin and soft tissue infection (15.2%). Severe infectious episodes were reported in 48.1% of the patients. The most frequent type of serious infections was pneumonia (26.6%) and neutropenic fever (13.9%). Nine patients (11.4%) died due to infections, including four from pneumonia, two from sepsis, one from neuroinfection, one from peritonitis, and one from diarrhea. Twenty-four of the patients (30.4%) did not experience an infectious episode during AZA treatment. One and two infectious complications were documented in 27.9% and 30.4% of the patients, respectively. At least four infectious episodes occurred in nine (11.4%) patients. The incidence rate of infectious complications in MDS patients treated with AZA is shown in Appendix A. 

Overall, 36 (45.6%) patients required at least one schedule modification or dose adjustment during AZA treatment (Appendix A). Dose reduction and treatment delays were documented in 23 (29.1%) and 7 (8.9%) patients, respectively. At least one episode of AZA cycle reduction (<7 days) was performed in 25 (31.6%) patients. 

### 3.4. Overall Survival and Prognostic Factors

Median OS was 17.6 months (95% CI, 13.56–23.83) (Figure 1A) and median PFS was 14.96 months (95% CI, 9.1–17.1) (Figure 1B). Patients with a hematological response (CR, PR or HI) and SD had a significantly longer median OS (23.8 vs. 5.7 months, *p* = 0.0005) and PFS (19.8 vs. 3.5 months, *p* < 0.001) compared to non-responders (Appendix A). 

In univariate analysis, only IPSS cytogenetic risk was significantly associated with response to AZA (*p* = 0.03). Patients with good and intermediate cytogenetic risk according to IPSS achieved a significantly higher rate of response (ORR or SD) than those with poor cytogenetic risk (Appendix A). There was no significant difference in response to any other analyzed clinical feature (Appendix A).

In univariate analysis the following clinical factors had a significant effect on OS: IPSS risk group (*p* = 0.001), IPSS-R risk group (*p* = 0.01), IPSS cytogenetic risk (*p* = 0.0014), IPSS-R cytogenetic risk (*p* = 0.0023), complex karyotype (*p* = 0.003), and monosomal karyotype (*p* < 0.001) (Table 4; Appendix A). In univariate analysis, age (*p* = 0.0099), IPSS risk group (*p* = 0.0005), IPSS-R risk group (*p* = 0.0069), IPSS cytogenetic risk (*p* = 0.016), IPSS-R cytogenetic risk (*p* = 0.012), complex karyotype (*p* = 0.017), monosomal karyotype (*p* = 0.006), and SF level (*p* = 0.007) were prognostic factors for PFS (Table 4). Patients with IPSS intermediate-1 and intermediate-2 risk achieved significantly longer OS and PFS than those with IPSS high risk. The analysis confirmed that IPSS-R score is a significant predictor of OS and PFS. The median OS and PFS of patients with poor cytogenetic risk according to IPSS was significantly shorter compared to the rest of the patients. Moreover, it was shown that IPSS-R cytogenetic risk stratification influenced OS and PFS. Patients with complex karyotype reached significantly inferior OS (12.67 vs. 24.33 months, *p* = 0.003) and PFS (11.38 vs. 17.1 months, *p* = 0.017) compared to those with noncomplex karyotype. Monosomal karyotype was associated with significantly shorter OS (6.8 vs. 24.08 months, *p* < 0.001) and PFS (6.8 vs. 17.1, *p* = 0.006). Patients younger than age 65 achieved significantly superior PFS (24.4 months) compared to older patients (*p* = 0.0099). 

In multivariate analysis, older age (*p* = 0.047), higher IPSS risk (*p* = 0.014), and higher IPSS-R cytogenetic risk (*p* = 0.004) were associated with significantly shorter OS in the study group. Furthermore, age (*p* = 0.001), IPSS risk (*p* = 0.02), IPSS cytogenetic risk (*p* = 0.002), and SF level (as a continuous variable, *p* = 0.008) were independent predictors of PFS in multivariate analysis. The results of the multivariate analysis are presented in Table 5.

## 4. Discussion

The development and successful implementation of AZA is considered a turning point in the history of MDS treatment. AZA was the first hypomethylating agent that changed the natural history of the disease and became a standard of care for higher-risk MDS patients. Due to DNA hypomethylation, AZA restores the expression of silenced tumor suppressor genes and exerts antitumor activity. The earliest evidence of the efficacy and safety of AZA comes from randomized clinical studies by the Cancer and Leukemia Group B (CALBG) and the AZA-001 trial. The three CALBG studies demonstrated a response rate of 40–47%, with CR rate of 10–17%. Furthermore, patients treated with AZA had significantly longer OS and time to AML progression compared to patients treated with best supportive care (21 vs. 13 months). In the AZA-001 study, 29% of patients achieved CR or PR. Moreover, treatment with AZA significantly improved OS compared to conventional care, including intensive chemotherapy (24.5 vs. 15 months) [2,3,4]. 

Nevertheless, the above-mentioned data come from clinical trials involving participants different from real-life patients. In daily clinical practice, physicians regularly face more complicated cases of patients being more likely to have poor performance status and many comorbidities. Therefore, it seems important to perform studies based on real-world evidence. In this single-center, retrospective analysis, we evaluated MDS patients treated with AZA in a real-life setting. For the analyzed group, the ORR was 24% with a CR rate of 13.9% and the median OS was 17.6 months. These results differ from previous real-world studies, particularly those conducted by French, Canadian, and Dutch Groups, in which patients reached higher ORR compared to the present analysis (43–48% vs. 24%) [7,8,9]. The difference in response rate between our analysis and the above-mentioned studies may be caused by the higher rate of HI in the latter (15–25% vs. 7%). Of note, in the referenced studies the CR rate (14–17%) was comparable to our analysis. Interestingly, the achievement of high ORR did not translate into an OS improvement, as the median OS was certainly shorter compared with the median OS achieved in our study (13–13.5 vs. 17.6 months). This is probably related to older age (71–74 vs. 68 years) and a higher rate of patients with unfavorable cytogenetic risk (38–49% vs. 34%). On the other hand, in another real-life AZA study, Helbig et al. [28] demonstrated very similar results to our study, in particular for ORR (27%). It should be underlined that both studies included a very similar population of patients in terms of age and cytogenetic risk.

The AZA 001 trial demonstrated that the achievement of response is associated with a significant increase in OS. Moreover, an improvement in survival was noted in all responders, which proves that CR is not mandatory for benefitting from AZA therapy [4]. Further analysis revealed that patients with SD also had notably longer survival, with a 95% decreased risk of death [29]. In our study, we confirmed that both responders and patients with SD have a better prognosis with significantly longer OS (23.8 vs. 5.7 months) and PFS (19.8 vs. 3.5 months) compared to non-responders. It should be noted that patients with SD were the most numerous group (40.5%) of MDS patients. These results comply with previously published data. For instance, in the Belgian study, OS was significantly longer in the responders’ group and in patients with SD compared with non-responders (16 vs. 6 months) [30]. Comparable results were reported by the Austrian, Dutch, and Portuguese groups [9,31,32].

In order to achieve a response to AZA therapy, it is of high importance to administer at least several courses of treatment, which is related to the necessity of reactivation of suppressor genes. In the current study, the median number of AZA cycles required to achieve the best response was six. In the CALBG 9211 trial, 75% of responses were obtained within the first four AZA cycles and 90% during the first six cycles [3]. Similarly, in the AZA-001 study, 81% of responses were documented after six cycles, and 90% after nine cycles of AZA [4]. In accordance with these results, in the majority of real-life studies, the median number of administered AZA cycles was 5–6 [7,8,9,28,33]. Additionally, in the AZA-001 trial, continuation of AZA treatment led to an improvement in response in 43% of patients [4]. Therefore, a response evaluation should be performed after a minimum of six courses of AZA [34]. Unfortunately, DNA hypomethylation during AZA therapy is transient and reversible. Therefore, treatment interruptions or premature discontinuation are associated with rapid progression of the disease. For this reason, it is recommended to continue AZA treatment for as long as patients experience clinical benefits [10].

In our study, unfavorable cytogenetic risk was the strongest negative prognostic factor in patients with MDS treated with AZA. Unfavorable changes in karyotype influenced both significantly lower response rates and shorter OS and PFS. Patients with good and intermediate cytogenetic risk according to IPSS achieved significantly higher rates of response and SD (73% and 88.9%, respectively) than patients with poor cytogenetic risk (48.1%). Additionally, the median OS and PFS of patients with unfavorable cytogenetic risk according to IPSS and IPSS-R was significantly inferior in comparison to the rest of the patients. Of note, the prognostic impact of cytogenetic risk was retained in the multivariate analysis. The prognostic value and clinical meaning of cytogenetic abnormalities in MDS patients treated with AZA were confirmed in several studies [7,8,9,33,35,36]. Special attention should be paid to the study by Itzykon et al. [8], in which poor IPSS cytogenetic risk was one of the independent predictors of shorter OS and was included in the prognostic scoring system. Similarly, a prognostic impact of IPSS-R cytogenetic risk was confirmed in the large study (n = 282) by the GFM Group [35]. The median OS for the good, intermediate, poor and very poor cytogenetics groups reached 21.8, 12.3, 15.1, and 7.1 months, respectively. By comparison, in Cluzeau et al.’s study, the IPPS-R cytogenetic risk had no impact on response and survival [37]. 

Complex and monosomal karyotypes are widely known to be the strongest negative prognostic factors in MDS patients. Their presence is associated with a high risk of transformation into AML and a very poor prognosis. In the study by Itzykon et al., patients with complex karyotype treated with AZA achieved a significantly shorter duration of response (4.6 vs. 10.3 months) compared to the rest of the patients. Moreover, patients with complex karyotype had a lower response rate (39% vs. 51%) [8]. Hwang et al. [38] confirmed in a multivariate analysis that complex karyotype is an independent factor of lower response rate and shorter OS in MDS patients treated with AZA. Our findings are complementary to these studies, as patients with complex karyotype achieved significantly shorter OS (12.7 vs. 24.3 months) and PFS (11.4 vs. 17.1 months) in comparison to the rest of the patients. In our study, the presence of a monosomal karyotype was associated with the shortest survival—the median OS and PFS reached only 6.8 months versus 24.1 and 17.1 months in the rest of the patients. Similar results were documented in the study by Cluzeau et al., in which MDS patients with monosomal karyotype had the most adverse outcome with a median OS of 9 months [37].

Patient age is one of the most important factors impacting the decision regarding treatment intensity. Elderly patients are more likely to achieve worse outcomes than younger patients due to poorer performance status, comorbidities, and increased toxicity of chemotherapy. The results of most studies indicate that AZA has the same efficacy in older patients [4,7,8,39]. In our study, patients aged <65 years achieved significantly longer PFS (24.4 months) in comparison with patients aged 64–74 and >75 years (12.5 and 16.0 months, respectively). Moreover, patient age was an independent factor of OS and PFS in multivariate analysis. 

Iron overload is often a clinical problem in transfusion-dependent patients, limiting the efficacy of MDS treatment. Secondary hemochromatosis may result in a higher risk of infections, multiple organ failure, and poor outcomes [12,40,41]. It has been confirmed that patients treated with AZA with higher SF levels achieve a significantly lower response rate and shorter survival [12,41]. The present study, based on univariate and multivariate analysis, revealed a negative prognostic value of higher SF level in terms of PFS. Patients with lower SF levels (<500 ng/mL and <750 mg/L) achieved significantly longer median PFS (15.0 and 15.4 months) than patients with higher SF levels (750–1000 ng/mL and >2000 ng/mL), who had the shortest median PFS at 4.4 and 4.9 months, respectively. Of note, patients with SF levels of 1000–2000 ng/mL reached the longest median PFS (21.8 months), but this result should be interpreted with caution due to the small number of patients (n = 8). 

AZA treatment was safe and well-tolerated. The frequency of adverse reactions was consistent with previous studies [28,30]. The most common AE were infections, injection site reactions, and neutropenia. The majority of non-hematological AE were mild and reported during the early stage of treatment. Hematological toxicity and concomitant infections were the main reasons for schedule modifications and dose adjustments during the AZA treatment. In the literature, the frequency of infections during AZA therapy varies between 8% and 71% [28,30,42,43,44,45]. In our study, infections were documented in 69.6% of patients, with two or more episodes in 41.8% of patients. As in previous studies, pneumonia and neutropenic fever were the most dominant serious infectious complications. 

The main limitation of our study was the lack of analysis of the presence of genetic mutations and their prognostic impact on response and survival. However, the study was conducted at a time when molecular testing was not the standard for MDS diagnosis, hence molecular examination was only performed in 14 patients (17.8%) from our cohort. Additional limitations of our study are due to the retrospective character of the analysis and the relatively small group of patients (n = 79). The benefit of our study is the fact that it was performed in a single center characterized by a unified standard of care.

## 5. Conclusions

In conclusion, we demonstrated that the achievement of remission is not a necessary condition for providing a survival benefit for MDS patients treated with AZA. Unfavorable cytogenetics risk is the strongest negative prognostic factor in MDS patients, which negatively influences response rate as well as OS and PFS. Furthermore, older age, a complex or monosomal karyotype, and a higher level of SF were additional unfavorable prognostic factors for OS and PFS. Moreover, we confirmed the prognostic value of the IPSS and IPSS-R scoring systems in MDS patients receiving AZA in daily clinical practice. However, further prospective studies are required to establish prognostic factors in MDS patients treated with AZA.

## Figures and Tables

**Figure 1 cancers-16-01333-f001:**
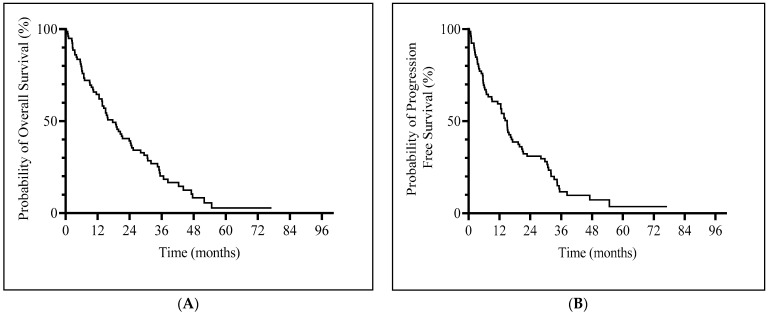
Kaplan–Meier curves of survival. (**A**) Overall survival. (**B**) Progression-free survival.

**Table 1 cancers-16-01333-t001:** Baseline patient characteristics (n = 79).

Parameter	Value
Median age (range) in years	69 (42–89)
Gender	
Female	34 (43%)
Male	45 (57%)
WHO diagnosis	
MDS-MLD	6 (7.6%)
MDS-RS	1 (1.3%)
MDS-EB1	26 (32.9%)
MDS-EB2	40 (50.6%)
MDS 5q−	1 (1.3%)
t-MDS	5 (6.3%)
WBC (G/L)	
Median (range)	2.25 (0.1–17.7)
ANC (G/L)	
Median (range)	0.88 (0.01–10.8)
PLT (G/L)	
Median (range)	63 (7–343)
HGB (g/dL)	
Median (range)	7.8 (4.2–13.8)
LDH (U/L)	
Median (range)	402.5 (162–1426)
Serum ferritin (ng/mL)	
Median (range)	480 (12.3–10,410)
Transfusion dependence	
RBC-TD	42 (53.2%)
RBC-TD + PLT-TD	14 (17.7%)
Transfusion independent	23 (29.1%)
Bone marrow blasts	
Median (range)	9.9 (0.6–19)
Peripheral blood blasts	
Present	28 (35.4%)
Absent	42 (53.2%)
Unknown	9 (11.4%)
IPSS risk group	
Intermediate-1	20 (25.3%)
Intermediate-2	45 (57%)
High	14 (17.7%)
IPSS-R risk group	
Intermediate-2	16 (20.3%)
High	29 (36.7%)
Very high	34 (43%)
Cytogenetic risk (IPSS)	
Good	37 (46.8%)
Intermediate	9 (11.4%)
Poor	27 (34.2%)
Unknown	6 (7.6%)
Cytogenetic risk (IPSS-R)	
Very good	1 (1.3%)
Good	36 (45.6%)
Intermediate	9 (11.4%)
Poor	6 (7.6%)
Very poor	21 (26.6%)
Unknown	6 (7.6%)
Complex karyotype	
Yes	22 (27.9%)
No	51 (64.6%)
Unknown	6 (7.6%)
Monosomal karyotype	
Yes	17 (21.5%)
No	56 (70.9%)
Unknown	6 (7.6%)
HCT-CI	
Median (range)	1 (0–7)
BMI	25.7
Median (range)	(18.0–36.1)

WHO: World Health Organization; MDS: myelodysplastic syndrome; MDS-MLD: myelodysplastic syndrome with multilineage dysplasia; MDS-RS: myelodysplastic syndrome with ring sideroblasts; MDS-EB: myelodysplastic syndrome with excess of blasts; U: unclassifiable; t-MDS: therapy-related myelodysplastic syndrome; WBC: white blood cell; ANC: absolute neutrophil count; PLT: platelets; HGB: hemoglobin; LDH: lactate dehydrogenase; IPSS: International Prognostic Scoring System; IPSS-R: International Prognostic Scoring System revised; HCT-CI: hematopoietic cell transplantation-specific comorbidity index; BMI: body mass index.

**Table 2 cancers-16-01333-t002:** Azacitidine treatment.

Parameter	Value
Time to treatment onset (months)	
Median (range)	1 (0–69)
Number of cycles	
Median (range)	6 (1–34)
Treatment duration (months)	
Median (range)	6 (1–37)
Time to best response (AZA cycles)	
Median (range)	6 (1–12)
Response status, n (%)	
ORR (CR + PR + HI)	19 (24%)
CR	11 (13.9%)
PR	2 (2.5%)
HI	6 (7.6%)
SD	32 (40.5%)
PD	16 (20.3%)
Unknown	12 (15.2%)
HI, n (%)	6 (7.6%)
HI-E	5 (6.3%)
HI-P	2 (2.5%)
HI-N	1 (1.3%)
Transfusion independence	
Yes	14 (25%)
No	42 (75%)

ORR: overall response rate; CR: complete remission; PR: partial remission; HI: hematologic improvement; SD: stable disease; PD: progressive disease; HI-E: hematologic improvement–erythroid; HI-P: hematologic improvement–platelet; HI-N: hematologic improvement–neutrophil.

**Table 3 cancers-16-01333-t003:** Hematological and non-hematological toxicity.

Variable, n (%)	Total Events	Grade 1–2	Grade 3–4
Hematological toxicity	28 (35.4%)	3 (3.8%)	25 (31.2%)
Neutropenia	16 (20.3%)	0 (0%)	16 (20.3%)
Thrombocytopenia	9 (11.4%)	2 (2.5%)	7 (8.9%)
Anemia	3 (3.8%)	1 (1.3%)	2 (2.5%)
Non-hematological toxicity	30 (38%)	23 (29.1%)	7 (8.9%)
Injection site reaction	17 (21.5%)	17 (21.5%)	0 (0%)
Gastrointestinal	6 (7.6%)	6 (7.6%)	0 (0%)
Bleeding events	6 (7.6%)	2 (2.5%)	4 (5.1%)
Heart arrhythmias	4 (5.1%)	4 (5.1%)	0 (0%)
Acute kidney injury	3 (3.8%)	3 (3.8%)	0 (0%)
Transaminases increase	2 (2.5%)	0 (0%)	2 (2.5%)
Other	3 (3.8%)	1 (1.3%)	2 (2.5%)
Infectious complications	Total events		Grade 3–5
Total	55 (69.6%)		38 (48.1%)
Pneumonia	23 (29.1%)		21 (26.6%)
Upper respiratory tract infection	15 (19%)		5 (6.3%)
Neutropenic fever	14 (17.7%)		11 (13.9%)
Skin and soft tissue infection	12 (15.2%)		5 (6.3%)
Diarrhea	8 (10.1%)		3 (3.8%)
Urinary tract infection	7 (8.9%)		3 (3.8%)
Sepsis	3 (3.8%)		3 (3.8%)
Bacteriemia	2 (2.5%)		0 (0%)
Other	10 (12.7%)		5 (6.3%)

**Table 4 cancers-16-01333-t004:** Univariate analysis of factors influencing survival.

Parameter	OS			PFS		
	Median (Months)	95% CI	*p*	Median (Months)	95% CI	*p*
Gender			0.29			0.36
Male	17.6	11.5–28.1		15.0	6.8–28.1	
Female	17.3	10.2–23.8		13.6	6.1–19.8	
Age			0.099			0.0099
<65	28.1	19–38.3		24.4	9.0–34.5	
65–74	13.6	9.0–15.7		12.5	5.6–15.0	
≥75	17.9	3.4–30.6		16.0	2.5–22.8	
WHO diagnosis			0.89			0.96
MDS-MLD	24.7	3.4–NE		18.8	3.4–NE	
MDS-RS	15.7	NE		15.7	NE	
MDS EB-1	14.8	10.2–23.8		13.3	5.7–21.0	
MDS-EB-2	19.4	9.0–25.3		12.8	6.3–20.7	
MDS 5q−	18.8	NE		17.1	NE	
t-MDS	15.8	4.0–NE		15.1	4.0–NE	
IPSS risk group			0.001			0.0005
Intermediate-1	30.6	12.5–35.2		29.7	12.5–34.5	
Intermediate-2	19.0	11.5–25.3		15.1	7.7–20.7	
High	8.6	5.5–15.0		6.6	1.0–12.5	
IPSS-R risk group			0.01			0.0069
Intermediate	25.0	10.2–35.2		23.4	6.1–32.0	
High	28.1	14.3–35.4		22.8	13.8–32.0	
Very high	12.0	5.8–15.8		7.3	3.9–13.8	
Cytogenetic risk (IPSS)			0.0014			0.016
Good	30.6	19.0–35.4		29.7	13.8–32.0	
Intermediate	19.8	0.3–NE		12.5	0.3–NE	
Poor	11.5	5.6–15.7		6.8	3.4–15.0	
Cytogenetic risk (IPSS-R)			0.0023			0.012
Very good	36.7	NE		32.0	NE	
Good	29.4	18.8–35.4		28.1	13.8–33.3	
Intermediate	23.8	5.5–NE		7.0	3.9–NE	
Poor	6.2	0.3–NE		6.0	0.3–NE	
Very poor	13.6	3.4–15.7		11.3	3.3–15.1	
Complex karyotype			0.003			0.017
Yes	12.7	3.4–15.8		11.4	3.3–15.2	
No	24.3	13.8–35.0		17.1	11.3–31.0	
Monosomal karyotype			<0.001			0.006
Yes	6.8	2.6–15.7		6.8	1.1–15.1	
No	24.0	15.0–32.0		17.1	12.5–29.7	
Peripheral blood blasts			0.71			0.54
Present	18.9	9.1–24.3		11.0	5.7–17.1	
Absent	20.4	13.8–30.6		15.3	12.8–281	
BMI			0.49			0.36
18.50<	19.3	NE		12.8	NE	
18.50–24.99	20.2	10.2–34.4		16.7	6.3–29.7	
25–29.99	18.2	13.6–28.1		15.2	9.1–28.1	
>30	12.5	6.1–19.0		7.7	3.3–13.8	
Transfusion dependence			0.54			0.6
RBC-TD	14.6	10.4–21.0		13.8	7.0–17.1	
RBC-TD + PLT-TD	14.1	2.3–38.3		14.1	2.3–38.3	
TI	24.3	15.0–30.8		16.3	6.0–30.6	
Serum ferritin (ng/mL)			0.07			0.007
<500	19.3	9.7–28.1		15.0	6.8–21.3	
500–750	20.1	2.5–42.3		15.4	2.5–31.0	
750–1000	6.4	0.8–18.8		4.4	0.8–17.1	
1000–2000	24.4	5.6–NE		21.8	5.6.–NE	
>2000	5.7	0.3–NE		4.9	0.3–NE	

OS: overall survival; PFS: progression-free survival; CI: confidence interval; NE: not estimated; WHO: World Health Organization; MDS: myelodysplastic syndrome; MDS-MLD: myelodysplastic syndrome with multilineage dysplasia; MDS-RS: myelodysplastic syndrome with ring sideroblasts; MDS-EB: myelodysplastic syndrome with excess of blasts; t-MDS: therapy-related myelodysplastic syndrome; IPSS: International Prognostic Scoring System; IPSS-R: International Prognostic Scoring System revised; BMI: body mass index; RBC: red blood cell; TD: transfusion dependent; PLT: platelets; TI: transfusion independence.

**Table 5 cancers-16-01333-t005:** Multivariate analysis of prognostic factors significantly influencing OS and PFS.

Parameter	PFS		
	HR	95% CI	*p*
Age (years)	1.05	1.02–1.08	0.001
IPSS risk	1.77	1.08–2.9	0.02
Cytogenetic risk (IPSS)	1.76	1.23–2.51	0.002
Serum ferritin (ng/mL)	1.00025	1.00006–1.0004	0.008
	**OS**		
Age (years)	1.02	1–1.05	0.047
IPSS risk	1.71	1.11–2.63	0.014
Cytogenetic risk (IPSS-R)	1.41	1.12–1.78	0.004

OS: overall survival; HR: hazard ratio; CI: confidence interval; IPSS: International Prognostic Scoring System; PFS: progression-free survival; IPSS-R: International Prognostic Scoring System revised.

## Data Availability

The data presented in this study are available on request from the corresponding author.

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
