# Peer review of "Real-World Outcome and Prognostic Factors in MDS Patients Treated with Azacitidine—A Retrospective Analysis"

_cancers, 2024, doi:10.3390/cancers16071333_

Round 1

Reviewer 1 Report

Comments and Suggestions for Authors

Authors from Poland comducted a great study with retrospective analysis included 79 patients with MDS treated with Azacitidine as first-line therapy in a real-life setting. They  confirmed the efficacy and safety of AZA in a real-world population.

However, some points should be clarified.

Starting with a minor issue, tables and figures should be inserted into the manuscript . Authors present tables and figures as supplementary files, but I think, they mean, that these tables and figures should become a part of the manuscript.  In MDPI journals, they should be in the manuscript.

When I look at KM curves, I do not have an impression that AZA therapy is associated with longer OS. Do I misunderstood results? If so, authors should better explain them.

All patients die within 5 years. Is it the true. No one survive?  Even in pancreatic cancer, 5-10% survice after 5 years.  I would think, MDS patents should too.

I would also recommend authors to check epidemiological terms. They write about protective factors doing retrospective study. Epidemiologist know that in such studies neither risk nor protective factors can be displayed but only associations.

Reviewer 2 Report

Comments and Suggestions for Authors

The study evaluated prognostic factors for AZA-treated MDS patients based on a retrospective analysis in a local MDS cohort. The study design is straightforward and analyses are easy to follow. The study's main limitation, as the authors also mentioned in the Discussion section, is lacking the mutational profile, a key factor for prognosis analysis in many treatments. However, I understand the data were collected very early on and would not request extra add-ons.

Comments on the Quality of English Language

Overall quality is fine. Language, wording, and layout should be polished further. 

Round 2

Reviewer 1 Report

Comments and Suggestions for Authors

Dear authors, thank you for adressing my comments.

I have still one point:

Response 4: Following your recommendation, we have checked epidemiological terms. We have not used term “protective factors”. In our work we used the term “Prognostic factors” meaning a characteristic that is associated with clinical outcome (survival). We have replaced “predictive” into “prognostic” factors in abstract section.

As epidemiologist I can say that there is no matter if you use prognostic or risk factor as terms.  My comment was about the difference between retrospective and prospective research design.  Your study is retrospective (not a randomized clinical trial). Such studies - I haveb been conducting several studies like that-  do not allow the conclusions about any risk or prognostic factors, as both terms mean a causal realtionship.  Such studies investigate and show associations with increased or descreased risk, mortality, survival and so on. You can also write positive or negative associations.

I mark now this coment as minor revision.
